# Selenite Downregulates STAT3 Expression and Provokes Lymphocytosis in the Liver of Chronically Exposed Syrian Golden Hamsters

**DOI:** 10.3390/molecules26185614

**Published:** 2021-09-16

**Authors:** María Elena Camacho-Moll, Adriana Sampayo-Reyes, Fabiola Castorena-Torres, Gerardo Lozano-Garza, Gabriela Alarcón-Galván, Alba Hernández, Ricard Marcos, Juan Manuel Alcocer-González, Reyes Tamez-Guerra, Mario Bermúdez de León

**Affiliations:** 1Centro de Investigación Biomédica del Noreste, Departamento de Biología Molecular, Instituto Mexicano del Seguro Social, Monterrey 64720, Nuevo León, Mexico; maria.camachomo@imss.gob.mx (M.E.C.-M.); hglgarza@hotmail.com (G.L.-G.); 2Departamento de Ciencias Básicas, Vicerrectoría de Ciencias de la Salud, Universidad de Monterrey, San Pedro Garza García 66238, Nuevo León, Mexico; gabriela.alarcon@udem.edu; 3Facultad de Ciencias Biológicas, Universidad Autónoma de Nuevo León, San Nicolás de los Garza 66455, Nuevo León, Mexico; a_sampayo@hotmail.com (A.S.-R.); juan.alcocerg@uanl.mx (J.M.A.-G.); reyes.tamezgr@uanl.edu.mx (R.T.-G.); 4Escuela de Medicina, Tecnologico de Monterrey, Monterrey 64710, Nuevo León, Mexico; fcastorena@tec.mx; 5Grup de Mutagènesi, Departament de Genètica i de Microbiologia, Facultat de Biociències, Universitat Autònoma de Barcelona, 08193 Cerdanyola del Vallès, Spain; alba.hernandez@uab.es (A.H.); ricard.marcos@uab.es (R.M.); 6Consortium for Biomedical Research in Epidemiology and Public Health (CIBERESP), Carlos III Institute of Health, 28029 Madrid, Spain

**Keywords:** arsenic, selenite, α-tocopherol, liver, hamster, oncogenes

## Abstract

Arsenic is considered a worldwide pollutant that can be present in drinking water. Arsenic exposure is associated with various diseases, including cancer. Antioxidants as selenite and α-tocopherol-succinate have been shown to modulate arsenic toxic effects. Since changes in *STAT3* and *PSMD10* gene expression have been associated with carcinogenesis, the aim of this study was to evaluate the effect of arsenic exposure and co-treatments with selenite or α-tocopherol-succinate on the expression of these genes, in the livers of chronically exposed Syrian golden hamsters. Animals were divided into six groups: (i) control, (ii) chronically treated with 100 ppm arsenic, (iii) treated with 6 ppm α-tocopherol-succinate (α-TOS), (iv) treated with 8.5 ppm selenite, (v) treated with arsenic + α-TOS, and (vi) treated with arsenic + selenite. Urine samples and livers were collected after 20 weeks of continuous exposure. The urine samples were analyzed for arsenic species by atomic absorption spectrophotometry, and real-time RT-qPCR analysis was performed for gene expression evaluation. A reduction in *STAT3* expression was observed in the selenite-treated group. No differences in *PSMD10* expression were found among groups. Histopathological analysis revealed hepatic lymphocytosis in selenite-treated animals. As a conclusion, long-term exposure to arsenic does not significantly alter the expression of *STAT3* and *PSMD10* oncogenes in the livers of hamsters; however, selenite down-regulates *STAT3* expression and provokes lymphocytosis.

## 1. Introduction

The toxic properties of arsenic have been known since the 13th century, and the use of arsenic as a poison has played a role in domestic and dynastic intrigues throughout history [1]. Depending on the level of exposure, inorganic arsenic can cause liver diseases, neurological disorders, cardiovascular lesions, cancer, among others [2]. In the human liver, chronic arsenic exposure has been associated with non-cirrhotic portal fibrosis, with or without the development of portal hypertension [3]. Since the introduction of arsenic to industrial processes, and due to anthropogenic activities, such as mining, the prevalence of several types of cancer has increased in workers in specific environments [2,4,5]. Arsenic is a well-known water pollutant that generates worldwide public health concerns.

In Mexico, aquifers represent a permanent source of water, and Mexican guideline value allows arsenic levels in drinking water below a concentration of 25 µg/L (25 parts per million, ppm); however, there are areas in Mexico where arsenic levels can reach 262.9 µg/L [6]. Furthermore, there is an association of skin cancer and diabetes mellitus prevalence in this arsenic chronically exposed areas [7]. The World Health Organization (WHO) has established a limit of 10 µg/L in drinking water and a provisional tolerable daily intake for inorganic arsenic (2.14 µg/kg/day) [8].

Arsenic compounds are mainly metabolized in the liver of humans and in most of the rodents; therefore, the liver is considered a major target of inorganic arsenic toxicity [9]. Furthermore, epidemiological studies have shown liver dysfunction due to chronic arsenic exposure such as the high prevalence of hepatomegaly [10,11,12]. However, the pathogenesis of liver injury in arsenicosis remains unclear; therefore, animal studies have been proposed in order to understand such pathogenesis. Because metabolism plays an important role in the effects associated with arsenic exposure, it is important to use animal models with maximum similarity to humans in terms of arsenic biotransformation, as it is the case of the hamster [13,14].

Some studies have mentioned that there are also cases where hepatocellular carcinoma (HCC) and hepatic cholangiosarcoma are associated with chronic arsenic exposure [15]. The theory states that HCC originates from liver cirrhosis, however, the histological changes observed in the liver after chronic arsenic exposure are mainly non-cirrhotic changes; however, fibrosis has also been included in HCC risk factors [16]. Portal fibrosis is part of the liver injury that occurs after arsenic exposure, in addition to other genetic and epigenetic modifications [17,18]. One of the genes which are actively involved in liver fibrogenesis and angiogenesis is Signal Transducer and Activator of Transcription 3 (*STAT3*) [19]. Furthermore, *STAT3* has been associated with liver carcinogenesis and cancer progression [20]. Recently it has been shown that the oncogene 26S Proteasome non-ATPase regulatory subunit 10 (*PSMD10*) induces *STAT3* activation in a tumor microenvironment [16]. Although arsenic can modify the expression of several genes [21,22], and some oncogenes such as *STAT3* and *PSMD10* have been associated with malignant transformation of hepatocytes [23,24,25,26], our knowledge of arsenic’s targets and effects on cancer etiology are unclear. Evidence has been accumulating in recent years about the important role that *PSMD10* plays in hepatocellular carcinoma, for instance it has been shown to prevent the degradation of Octamer-binding transcription factor 4 (Oct4), regulates Retinoblastoma 1 (Rb1), tumor protein 53 (Tp53) and cisplatin sensitivity. It is has been proposed as a novel therapeutic agent to a specific dose.

The co-administration of antioxidants such as alpha-tocopherol (α-TOS) or sodium selenite reduces the toxic effects of sodium arsenite [14]. Therefore, in the present study we aimed to investigate whether arsenic toxicity could be modulated by the co-treatment with the two indicated antioxidants through *STAT3* and *PSMD10* oncogene expression in the liver of Syrian golden hamsters.

## 2. Results

Hamsters in each group were monitored every week for 20 weeks to evaluate weight gain and survival, in animals exposed chronically to arsenic and/or the selected antioxidants (selenite and α-tocopherol). Arsenic dose was chosen based in our previous studies [14,21], where it resembles the arsenic toxicity effect observed in human population [8]. For α-TOS and selenite doses, they were chosen on previous studies in animal models where not toxicity was observed by these two antioxidants [14,21,27,28]. No difference in weight gain was found among treatments, except in selenite-treated hamsters at 10 weeks compared to the control group (*p* < 0.05, Figure 1A). Survival rates were not affected by the treatments (Figure 1B).

We next determined the arsenic species in the urine to evaluate whether exposure to the antioxidants could affect arsenic metabolism. The assessment of arsenic species in urine samples demonstrated higher concentrations of inorganic arsenic (iAs) and methylarsonous acid (Mas) in arsenic and arsenic + α-TOS exposed hamsters compared to controls (Figure 2A,B). DMAs differences were observed in the arsenic, arsenic + α-TOS, and arsenic + selenite exposed groups compared to controls (Figure 2C). Ratios of iAs to MAs concentrations were higher in arsenic- and selenite-treated animals compared to controls (Figure 2D), but no significant difference was observed in the ratios of MAs to dimethylarsinous acid (DMAs) (Figure 2E). In conclusion, the antioxidants tested (α-TOS and selenite) do not have a significant effect on arsenic metabolism.

To investigate whether chronic exposure to arsenic affects the expression levels of *STAT3* and *PSMD10* genes, the livers of the animals were collected and gene expression was evaluated. Although there was a trend towards *STAT3* downregulation in α-TOS, Arsenic + α -TOS- and Arsenic + Selenite-treated groups, only the *STAT3* expression was significantly reduced in selenite-treated animals compared to controls (Figure 3A). The arsenic-exposed group showed a *STAT3* upregulation trend, but it was not significant. Compared to controls, there were no statistically significant difference in *PSMD10* gene expression for the different exposure conditions (Figure 3B); however, a trend towards a decrease was observed in Selenite- and Arsenic + selenite-exposed animals.

Two random samples of each treatment were analyzed for microscopical alterations. Examination of hematoxilyn- and eosin-stained tissue samples demonstrated that, in general, all tissues had a degree of vascular congestion and steatosis (Appendix A). Only one of the two examined controls showed no alterations. Moderate sinusoidal lymphocytosis was present in selenite-treated samples (Figure 4A,B), and samples treated with arsenic + selenite had augmented lymphocytosis with lymphocyte aggregation (Figure 4C–F). Some of these aggregates had mature lymphocytosis and an active center with centroblasts (Figure 4C,D).

## 3. Discussion

Arsenic chronic exposure is an important issue in public health. In Mexico and other regions in the world, there are areas where the human population is environmentally exposed to arsenic in drinking water, and some studies have reported association of this chronic exposure and diseases [7,15,29,30]. Therefore, the present study aimed to determine whether arsenic toxicity could be modulated by selenite and α- tocopherol, two antioxidants which have been shown to ameliorate arsenic toxic effects [21,31,32], and we used to oncogenic markers, i.e., *STAT3* and *PSMD10* genes, and an histological approach to reveal the grade of liver alteration.

Arsenic is ingested as inorganic arsenic, which is the metabolized through reduction-oxidation and methylation reactions. DMAIII and DMAV are eliminated in urine as demethylated forms, followed by the monomethylated forms MMAIII and MMAV. It has been shown that there are sex differences in arsenic metabolism and toxicity and therefore only male hamsters were used in the present study [33,34,35,36,37]. It was first believed that arsenic methylation metabolites such as MAs and DMAs were less cytotoxic compared to iAs,; however, further studies demonstrated that DMAs was teratogen, acting as a nephrotoxin, a tumor promoter and carcinogenic in the rat [38]. Our data demonstrate that arsenic was converted to its DMA and MAs form, as observed in urine samples. However, the increased levels of DMAs observed in the arsenic exposed group were not reduced by the treatment with antioxidants. These results disagree with previous studies showing that α-TOS does affect iAs and DMAs levels in patients at risk of urothelial carcinoma, and in hamsters [15,39]. This study demonstrated that chronic exposure to arsenic at a dose of 100 parts per million (ppm), that resembles the arsenicosis observed in human population [14], does not cause histologically detectable hepatic changes compared to controls in the animal model. Furthermore, the expression of *STAT3* and *PSMD10* genes was not found to be affected by arsenic exposure.

Selenium is an antioxidant, which can cause toxic effects when administered slightly above its homeostatic required levels. Bodyweight reduction and an immunological response have been associated with selenite intoxication at 9 ppm [40,41]. However, the doses used in the present study are well below the daily permissible doses in hamsters [27,28]. Selenite exposure was found to affect *STAT3* gene expression. Additionally, a trend towards a decreased weight in selenite-exposed animals was observed, coinciding with the treatments associated with hepatic lymphocytosis. In a previous study, 10 mg/kg sodium selenite caused hepatic changes, including a focal accumulation of inflammatory cells [42] similar to those observed in this work. However, there does not appear to be any link between the observed hepatic changes and weight loss. As mentioned in results section, steatosis was observed even in control animals, perhaps due to diet. Moreover, selenite is a known anticarcinogen, but it can also trigger cancer [38,43]. Conversely, although arsenic is a well-known carcinogen, when it is administered as arsenic trioxide, it is able to reduce certain types of cancer in mice and human [44,45,46,47]. Hence, there are dual roles as anticancer and carcinogenic effects for selenium and arsenic [46]. We have shown that sodium arsenite reduces the effects of sodium selenite on *STAT3* expression. Thus, although selenite alone reduced *STAT3* expression, no effects were observed when combined with sodium arsenite.

On the other hand, it seems that sodium arsenite contributed to sodium selenite-induced lymphocytosis in the hamster livers. Studies in other species have shown that blood lymphocytosis is present when animals are exposed to low levels of selenium [48]. The observation of lymphocytosis in animals not exposed to selenium suggests that other factors, such as stress or infection, could be involved in such process. In the present study, lymphocytosis was observed mainly in selenium-exposed animals, with a greater effect when selenium was combined with arsenic, suggesting that arsenic would potentiate the selenium induced lymphocytosis. In mice, dietary selenium has shown to modify lymphocyte activation and differentiation through genetic modifications [49]. This effect has also been observed in humans, where selenium supplementation increased the expression of glutathione peroxidase homologs 1 and 4 (*GPX1* and *GPX4*, respectively) in lymphocytes. These two genes are involved in protecting lymphocytes from oxidative stress [50,51]. Because the xenobiotics to which we exposed the hamsters are metabolized in the liver, the presence of selenite in the liver could modify *STAT3* gene expression. Tsuji et al. reported pro-inflammatory liver response to selenium in mice, but they do not observe effects by arsenic exposure [31] as we have found. There are other studies reporting plant extracts with antioxidants to protect human cells from environmental oxidative stressors [52,53], which could be explored in the case of arsenic toxicity.

Further studies, such as those using immunohistochemistry and real-time qPCR for different lymphocyte markers, should be performed to assess this effect and to determine what type of lymphocytes are present in the liver given that *STAT3* is a central regulator of lymphocyte differentiation and function [54], and it is involved in the generation of inflammatory helper T cells [55]. Furthermore, given that interleukin-6 and interleukin-23 activate *STAT3* expression, interleukin expression should also be assessed [56,57].

## 4. Materials and Methods

### 4.1. Chemicals

Sodium arsenite (S-7400), D-α-tocopherol succinate (T-3126), and sodium selenite (S5261) were purchased from Sigma-Aldrich (St. Louis, MO, USA).

### 4.2. Animals and Treatments

This study was approved by the institutional ethics committee and conducted in accordance. The National Institutes of Health guide for the care and use of laboratory animals were followed. All procedures involving animals were conducted in accordance with the ethical standards of the institution. This study is registered under the number R-2010-1906-28.

Fifty-four Syrian golden male hamsters (*Mesocricetus auratus*) weighing 76.5–89 g were randomly divided and housed into six treatment groups, as summarized in Table 1. Food (5001 Rodent Diet, LabDiet, PMI^®^ Nutrition International, LLC, Brentwood, MO, USA) and water were provided ad libitum, and each group received the corresponding treatment regimen for 20 weeks before euthanasia. All compounds were administered via drinking water. Doses for the administered compounds are listed in Table 1. The sodium arsenite dose was based on previous studies [21,58], and the doses of α-tocopherol succinate (α-TOS) and sodium selenite were chosen based on studies showing no toxicity in animals [14,21,27,28]. Before euthanasia, animals were housed in metabolic cages. Total urine volume was collected for 24 h in a waste collector. Urine samples were frozen at −20 °C until analysis. Animals were euthanized using a CO_2_ chamber and cervical dislocation, followed by the collection of the liver. Livers were kept in RNAlater RNA Stabilization Solution (Invitrogen, Carlsbad, CA, USA) at −20 °C until ready for RNA extraction.

### 4.3. Measurement of Arsenic and Arsenic Species

The separation and quantification of arsenic species, i.e., inorganic arsenic (iAs), methylarsonous acid (MAsIII), methylarsonic acid (MAsV), dimethylarsinous acid (DMAsIII), and dimethylarsinic acid (DMAsV) and the trivalent and pentavalent forms, were assessed by the Laboratorio de Investigación y Servicios en Toxicología (LISTO-CINVESTAV) by hydride-generation atomic absorption spectrometry (HG-AAS), using cryotrapping (AS) as previously described [59]. Briefly, the system consists of a flow injection system, a computer, an arsenic electrodeless discharge lamp (Perkin Elmer, Waltham, MA, USA) that serves as a radiation source at 390 mA. For total arsines (total As, iAsIII + iAsV), MAs (MAsIII + MAsV) and DMAs (DMAsIII + DMAsV), samples were incubated with Cysteine hydrochloride (2% Cys and 0.11 M HCl final concentrations; pH 1.5) for 70 min at room temperature. Treatment with cysteine reduced all pentavalent As species to trivalency. After treating samples with Cys arsines were generated on the previously described system, where there was a gas–liquid separation where arsines were generated and deposited in the separator at a preset sample volume (0.025–0.8 mL), deionized water was then added to complete the 0.8 mL. The sample was then mixed with 1 mL NaBH_4_ and 1 mL Tris-HCl (0.75 M). The mixture reached a final pH of between 1 and 2 and arsines were formed. Arsines were then swept with helium (100 mL/min) and a gradient of temperature of −293 to 50 °C (this was achieved by the use of a cryotrap of liquid nitrogen and heat generated by an electric current applied on a Ni/Cr wire). Arsines were released at different temperatures iAs at −55 °C, MAs at 2 °C, and DMAs at 36 °C. The atomization of arsines was achieved by a microflame of hydrogen and air, with a flow of 23 and 42.9 mL/min, respectively. Arsines were detected with an atomic absorption spectrophotometer. The width of the measurement band was 0.7 nm and the background signal was corrected with a deuterium lamp. Signals were exported as ASCII files on the Origin Pro 7.5 (OriginLab corporation, Northampton, MA, USA) software.

### 4.4. RNA Extraction and cDNA Synthesis

RNA was extracted from a 50–100 mg liver piece from right dorso-caudal lobe, which was chopped with a scalpel and transferred into a 1.5 mL microtube containing 1 mL of TRIzol reagent (Invitrogen, Carlsbad, CA, USA). Samples were mixed manually by inversion for 10 min followed by the addition of 200 µL of chloroform (Tedia, Fairfield, OH, USA), mixed by inversion and incubated for 3 min at room temperature. Samples were then centrifuged for 15 min at 4 °C and 12,000× *g*. The aqueous phase was collected and transferred to a new tube. A total of 500 µL of isopropanol (Tedia) were added to the tube, mixed by inversion, and incubated at room temperature for 10 min. Samples were then centrifuged for 10 min at 4 °C and 12,000× *g*. The supernatant was discarded and the pellet was washed with 1 mL of 75% cold ethyl alcohol (Sigma-Aldrich, St. Louis, MO, USA). Samples were then mixed by inversion and centrifuged for 5 min at 4 °C at 7500× *g*. Supernatant and remaining ethyl alcohol were discarded; the rest was allowed to evaporate for 5–10 min at room temperature. The pellet was resuspended in 30 µL of nuclease-free water and stored at −70 °C.

Complementary DNA (cDNA) was synthesized by mixing 1 µL of random primers (ThemoFisher Scientific, Carlsbad, CA, USA) and 1 µL of dinucleotides (Invitrogen) with 10 µL of total RNA, at a final concentration of 2 ng/µL. Samples were loaded in a thermocycler (Veriti, Applied Biosystems, Foster City, CA, USA) and incubated for 5 min at 65 °C, followed by the addition of 4 µL of 5× first strand buffer (Invitrogen), 2 µL of dithiothreithol (Invitrogen), and 1 µL of RNase Out (Invitrogen). Samples were then incubated for 2 min at 37 °C and after this step 1 µL of M-MLV enzyme (Invitrogen) was added to the reaction. Samples were then incubated at 25 °C for 10 min, 37 °C for 50 min and finally 70 °C for 15 min. Samples were then stored at −20 °C until its analysis. The cDNA was tested by the amplification of the Gapdh gene.

### 4.5. SYBR Green Quantitative Real-Time Reverse Transcriptase (RT)-PCR

SYBR green RT-PCR was performed to determine *STAT3* and *PSMD10* relative expression in the livers of the animals. Primer sequences were STAT3 FWD 5′-GAG GCA TTC GGG AAG TAT TGT-3′, STAT3 RVS 3′-CAT CGG CAG GTC AAT GGT ATT-5′, PSMD10 FWD 5′-GAG ATT GTA AAA GCC CTT CTG-3′, PSMD10 RVS 3′-GAT TTG CCC CAC CTT CTA GT-5′, Gapdh FWD 5′- TCC TTG GAG GCC ATG TGG GCC AT-3′, Gapdh RVS 3′ CTT CAC CAC CTT CTT GAT GTC ATC A-5′. All primers were obtained from Integrated DNA Technologies (IDT, Skokie, IL, USA). SYBR green RT-PCR was performed using the SYBR green master mix as per manufacturer’s instructions (Applied Biosystems, Foster City, CA, USA). Real-time PCR was performed in an ABI 7500 Fast (Applied Biosystems) device, the program was set at 95 °C for 10 min, followed by 50 cycles of 95 °C for 5 secs and 60 °C for 1 min. Results were analyzed using the ΔΔCT method and relative expression to *Gapdh* gene was calculated.

### 4.6. Hematoxylin and Eosin Staining

Representative liver samples of each treatment were obtained and fixed in 4% formaldehyde followed by the processing and staining of the tissue for pathology analysis in an external laboratory (Centro de Patología Veterinaria in Guadalajara, Jalisco, Mexico; http://www.patvet.com.mx/ (accessed on 5 September 2021)). Images were taken on a Zeiss Primo Star educational microscope (Zeiss, Oberkochen, Germany).

### 4.7. Data Analysis

Data were analyzed using GraphPad Prism 6.04 (La Jolla, CA, USA). All data were tested for normality with a Shapiro–Wilk test. Animal survival analysis was performed with a survival curve comparison. Animal weight data are shown in relative units and analyzed with a two-way analysis of variance (ANOVA); Bonferroni tests were used for multiple comparisons. *STAT3* and *PSMD10* gene expression data were analyzed with an ordinary one-way ANOVA and Bonferroni tests for multiple comparisons. In non-normal distribution, PSMD10 data were analyzed with a non-parametric one-way ANOVA (Kruskal–Wallis test) due to a significant Shapiro-Wilk test, followed by a Dunn’s test for multiple comparisons.

## 5. Conclusions

As conclusion, long-term exposure to arsenic does not alter significantly the expression of *STAT3* and *PSMD10* oncogenes in the livers of hamsters; however, selenite down-regulates *STAT3* expression and provokes lymphocytosis in the liver. It is possible that the specific induction of genes involved in oxidative stress protection, such as *GPX1* and *GPX4*, in lymphocytes by selenite could increase its levels and aggregation in the tissue.

## Figures and Tables

**Figure 1 molecules-26-05614-f001:**
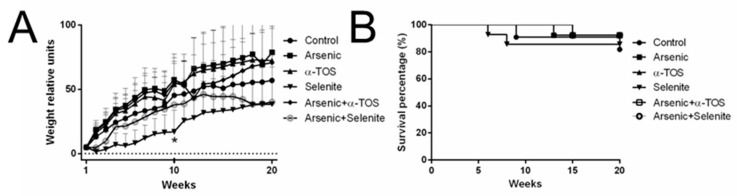
Weight gain and Kaplan–Meier survival analysis in arsenic-chronically exposed hamsters. Panel (**A**), weight relative units for control, arsenic, α-tocopherol succinate (α-TOS), selenite, arsenic + α-TOS, and arsenic + selenite exposed hamsters. Weight was recorded every week during 20 weeks. Data analyzed by two-way ANOVA and Bonferroni post-hoc test; means ± SD; *, *p* < 0.05. Panel (**B**), Kaplan–Meier curve was determined in 77 individuals over 20 weeks; means ± SD.

**Figure 2 molecules-26-05614-f002:**
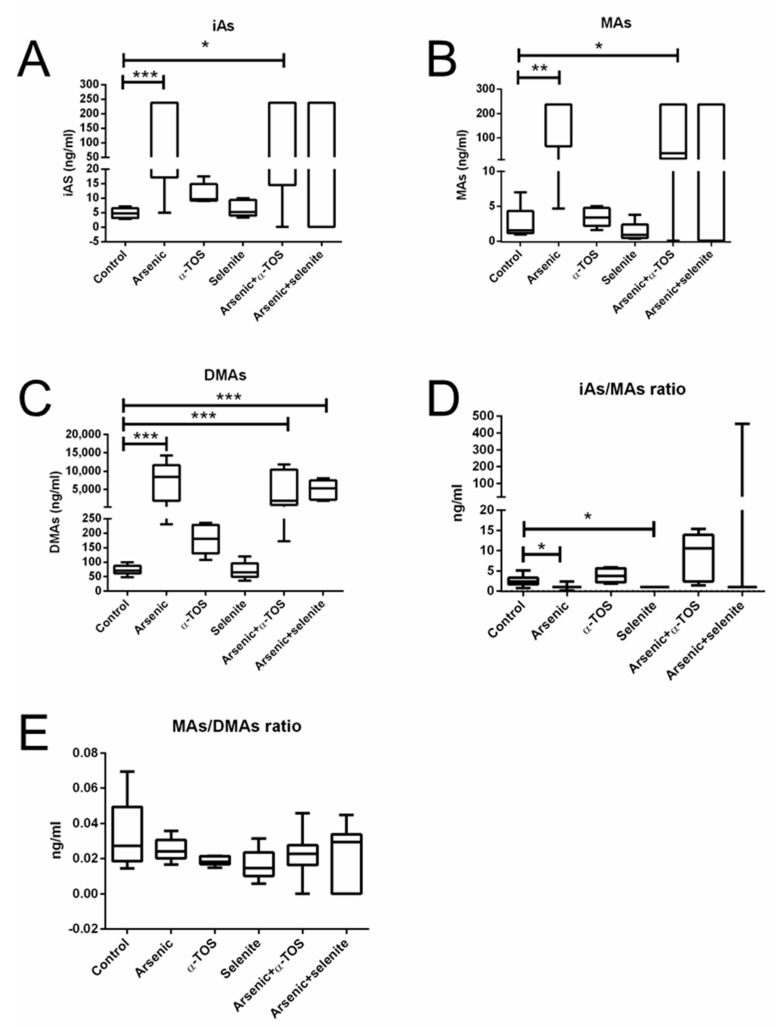
Box plots for the levels of arsenic species in the urine of arsenic-chronically exposed hamsters. Panels (**A**–**C**), concentrations of inorganic arsenic (iAs, Panel (**A**), methylarsonous acid (MAs) species (Panel (**B**)), and dimethylarsinous acid (DMAs) species (Panel (**C**)); Panel (**D**), iAs/MAs ratios; Panel (**E**), MAs/DMAs ratios. Data analyzed by Kruskal–Wallis test; means ± SD; *, *p* < 0.05; **, *p* < 0.01; ***, *p* < 0.001. The median is represented by the midline. The whiskers represent the minimum and maximum values.

**Figure 3 molecules-26-05614-f003:**
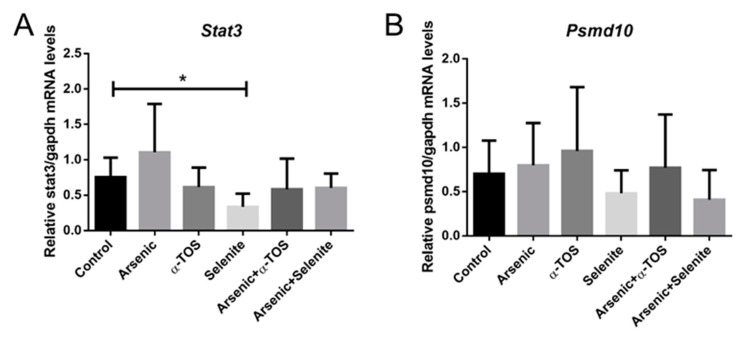
Relative levels of expression for *STAT3* (Panel (**A**)) and *PSMD10* (Panel (**B**)) genes in livers of Syrian golden hamsters chronically exposed to the different conditions. Relative values of mRNA were normalized by *Gapdh* expression in all samples. Data analyzed with Kruskal–Wallis test corrected for multiple comparison with Dunn test; means ± SD; *, *p* < 0.05.

**Figure 4 molecules-26-05614-f004:**
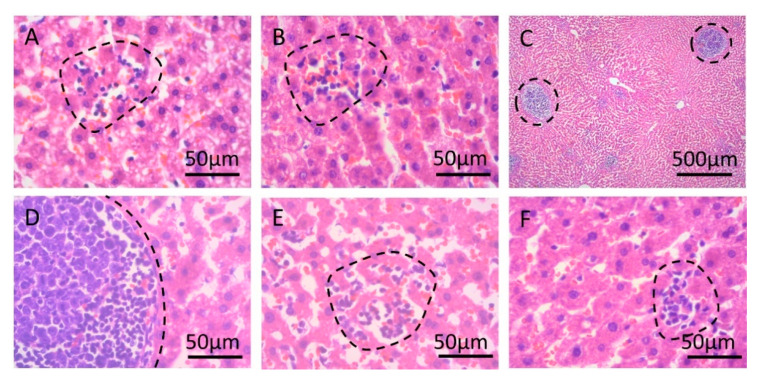
Representative images of hematoxylin- and eosin-stained sections from Syrian golden hamster livers chronically exposed to arsenic. Lymphocytosis was observed in specimens from two animals exposed to selenite (Panels (**A**,**B**)) and two exposed to arsenic + selenite (Panel (**C**–**E**) correspond to the first specimen, and (**F**) corresponds to the second specimen). Dashed lines represent areas with lymphocytosis.

**Table 1 molecules-26-05614-t001:** Summary of Group sizes, treatments, and doses used per treatment.

Group	*n*	Treatment
Control	9	Tap water
Arsenic	10	Sodium arsenite, 100 ppm
α-TOS	9	α-TOS, 6 ppm
Arsenic + α-TOS	9	Sodium arsenite and α-TOS
Selenite	10	Sodium selenite, 8.5 ppm
Arsenic + Selenite	7	Sodium arsenite and sodium selenite

α-TOS, α-tocopherol succinate.

## Data Availability

The data presented in this study are available on request from the corresponding author.

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
