# Peer review of "Selenite Downregulates STAT3 Expression and Provokes Lymphocytosis in the Liver of Chronically Exposed Syrian Golden Hamsters"

_molecules, 2021, doi:10.3390/molecules26185614_

Round 1
Reviewer 1 Report
The scientific paper by Camacho-Moll et al. entitled “The downregulation of the Stat3 gene induced by selenium, in 2 the liver of chronically exposed Syrian golden hamsters, is not 3 modified by arsenic co-exposure” aimed to evaluate the effect of arsenic exposure on the expression of the Stat3 and Psmd10 genes, in the 27 livers of Syrian golden hamsters, as well as investigate the effects of co-administered antioxidants. The methodology used in this research is appropriate for the hypotheses tested and the conclusions discussed are consistent with the experimental data. However, the manuscript could be improved by adding some clarifications.
Point 1: The used doses should be mentioned not only in the manuscript body, but also in the abstract (in the place where the experimental groups are listed) because the abstract should be able to stand on its own.
Point 2: The authors have stated that the sodium arsenite dose was chosen based on a previous study, while the doses of α -TOS and sodium selenite were chosen based on the results from the literature. However, it would be better if the dose selection was described in more detail in the manuscript body.
Point 3: In the ‘Measurement of arsenic and arsenic species’ section, which method was used for the sample preparation? This should be additionally explained. In this section, it should also be explained in which samples arsenic was measured (it was stated in the results that arsenic species were measured in the urine, but this should also be mentioned in the Material and methods section). Furthermore, the sentence “The assessment of arsenic species in 100 urine samples, performed as previously reported” in the results section is unnecessary, since this was also mentioned in the M&M.
Point 4: What was the reasoning for not using the same number of animals in all the experimental groups? Why were only male and not female hamsters used?
Point 5: Statistical tests should be mentioned in the legends of all the figures.
Point 6: The legend of the Figure 4 should contain a brief description of the effects observed on each picture. It would also be helpful if histological changes were marked on the pictures (i.e. by using arrows). In addition, frequency of effects should be given instead of statements such as ‘Moderate sinusoidal lymphocytosis was present in most of the selenite-treated species’.
Point 7: Although the discussion is structured well, it lacks the comparison of the results obtained in this study with the results from other similar studies present in the literature, on both arsenic toxicity observed in animal studies and selenium ameliorative effects.
Minor: In the Material and Methods section (animals and treatment), the authors should consider switching the word ‘sacrifice’ with ‘euthanasia’, since it is more appropriate.
Author Response
Comment:
The scientific paper by Camacho-Moll et al. entitled “The downregulation of the Stat3 gene induced by selenium, in 2 the liver of chronically exposed Syrian golden hamsters, is not 3 modified by arsenic co-exposure” aimed to evaluate the effect of arsenic exposure on the expression of the Stat3 and Psmd10 genes, in the 27 livers of Syrian golden hamsters, as well as investigate the effects of co-administered antioxidants. The methodology used in this research is appropriate for the hypotheses tested and the conclusions discussed are consistent with the experimental data. However, the manuscript could be improved by adding some clarifications.
The used doses should be mentioned not only in the manuscript body, but also in the abstract (in the place where the experimental groups are listed) because the abstract should be able to stand on its own.
Response: Thank you for your observation, doses are now included in the Abstract section.
Comment: The authors have stated that the sodium arsenite dose was chosen based on a previous study, while the doses of α -TOS and sodium selenite were chosen based on the results from the literature. However, it would be better if the dose selection was described in more detail in the manuscript body.
Response: We have described the dose selection on Results and Materials and Methods section. Also, we decided to refer to parts per million in all molecules for better comprehension (see Table 1).
Comment: In the ‘Measurement of arsenic and arsenic species’ section, which method was used for the sample preparation? This should be additionally explained. In this section, it should also be explained in which samples arsenic was measured (it was stated in the results that arsenic species were measured in the urine, but this should also be mentioned in the Material and methods section).
Response: Thank you for your comment. We have included more specific information about measurement of arsenic and arsenic species in Materials and Methods section.
Comment: Furthermore, the sentence “The assessment of arsenic species in 100 urine samples, performed as previously reported” in the results section is unnecessary, since this was also mentioned in the M&M.
Response: Thank you for your comment. The sentence has been removed.
Comment: What was the reasoning for not using the same number of animals in all the experimental groups? Why were only male and not female hamsters used?
Response: Regards to the number of animals per group, we distributed 10 animals per group and, before starting the treatments, some animals die in control, α-TOS, Arsenic+α-TOS and arsenic+selenite groups. Respect to the use of males, it has been shown that female rodents respond differently to arsenic exposure, for instance in utero exposure to arsenic has been shown to produce hepatocellular carcinoma in adult male offspring but not in female offspring [1], [2]. Due to the potential hormonal roles in arsenic metabolism, only males were used. Sex differences have also been observed in humans [3] where it has been shown that women have higher arsenic methylation efficiency than men, as observed on the reproductive age. Gut microbiome has also been shown to play a role in arsenic metabolism and there are also sex differences observed in this regard, demonstrating that female mice are more heavy metal resistant compared to male mice [4]. Finally, sex differences in arsenic metabolism has also been shown in humans [5]. All this has been better clarified in the Discussion section.
Comment: Statistical tests should be mentioned in the legends of all the figures.
Response: Thank you for your recommendation. Statistical tests have been added in legends.
Comment: The legend of the Figure 4 should contain a brief description of the effects observed on each picture. It would also be helpful if histological changes were marked on the pictures (i.e. by using arrows). In addition, frequency of effects should be given instead of statements such as ‘Moderate sinusoidal lymphocytosis was present in most of the selenite-treated species’.
Response: Thank you for your comment. As we mentioned in the Result sections and figure 4 legend, only two specimens of each treatment showed lymphocytosis. A dashed line delimiting the lymphocytosis was included in each photography to denote the event observed.
Comment: Although the discussion is structured well, it lacks the comparison of the results obtained in this study with the results from other similar studies present in the literature, on both arsenic toxicity observed in animal studies and selenium ameliorative effects.
Response: Thank you for your comment. Discussion section has been rewritten with more published studies.
Comment: In the Material and Methods section (animals and treatment), the authors should consider switching the word ‘sacrifice’ with ‘euthanasia’, since it is more appropriate.
Response: Thank you for your suggestion. We have changed the word
Reviewer 2 Report
The manuscript entitled “The downregulation of the Stat3 gene induced by selenium, in the liver of chronically exposed Syrian golden hamsters, is not modified by arsenic co-exposure” by Camacho-Moll et al. describes wide range of in vivo and biochemical studies in order to evaluate toxic effects of such know toxins as arsenic and inorganic selenium salts on liver including regulation of genes associated with tumor.
The research seems to be carried out very carefully as are all descriptions of the results and the experimental part. The observations made are logical and the discussions are based on the literature reports on previous research in this field.
However, Authors have analyzed an influence of various combinations of these poisons on weight of animals and other parameters during 20- week in vivo studies!
Although I appreciate scientific level of these studies, it is hard to me to justify the sense of such drastic research on animals carried out for the intended purpose.
Authors showed the approval of the relevant ethics committee but I do not find it right to torture 54 animals for half a year for the purpose of testing commonly known poisons. I do not think that the sentence "Mexican law allows arsenic levels in drinking water below a concentration of 25 µg / L" is a justification here.
Could Authors, please, indicate a more credible justification for such research.
Are the Authors going to use arsenic or inorganic selenium compounds in any important new therapy?
I believe that this work should be published because of valuable scientific observations and in order not to waste the suffering and death of these 54 animals.
Nevertheless, I strongly encourage Authors to use their interesting research models in vivo (presented in this paper) only to screen potential drugs that have a chance to save lives and thus in vivo assays could be really necessary. For purposes of understanding cellular or molecular mechanisms of obvious poisons, however, they should develop and use alternative in vitro models.
Author Response
Reviewer 2
Comment: The manuscript entitled “The downregulation of the Stat3 gene induced by selenium, in the liver of chronically exposed Syrian golden hamsters, is not modified by arsenic co-exposure” by Camacho-Moll et al. describes wide range of in vivo and biochemical studies in order to evaluate toxic effects of such know toxins as arsenic and inorganic selenium salts on liver including regulation of genes associated with tumor.
The research seems to be carried out very carefully as are all descriptions of the results and the experimental part. The observations made are logical and the discussions are based on the literature reports on previous research in this field.
However, Authors have analyzed an influence of various combinations of these poisons on weight of animals and other parameters during 20- week in vivo studies!
Although I appreciate scientific level of these studies, it is hard to me to justify the sense of such drastic research on animals carried out for the intended purpose.
Authors showed the approval of the relevant ethics committee but I do not find it right to torture 54 animals for half a year for the purpose of testing commonly known poisons. I do not think that the sentence “Mexican law allows arsenic levels in drinking water below a concentration of 25 µg / L” is a justification here.
Could Authors, please, indicate a more credible justification for such research.
Are the Authors going to use arsenic or inorganic selenium compounds in any important new therapy?
I believe that this work should be published because of valuable scientific observations and in order not to waste the suffering and death of these 54 animals.
Nevertheless, I strongly encourage Authors to use their interesting research models in vivo (presented in this paper) only to screen potential drugs that have a chance to save lives and thus in vivo assays could be really necessary. For purposes of understanding cellular or molecular mechanisms of obvious poisons, however, they should develop and use alternative in vitro models.
Response: Thank you very much for your comments and concerns about this study. The intention with our approach is to seek strategies to lessen the effect of arsenic using antioxidants in populations environmentally exposed to this metalloid. In some regions of our country, as well as in other parts of the world, arsenic represents a serious public health problem, and to propose effective strategies to reduce its harmful effects, it is necessary to use animal models that resemble human physiology, in this case, hamsters. In Mexico, there are several areas where people are constantly exposed to arsenic in drinking water and food. As we mentioned in the manuscript, Mexican law allows 25 µg/L of arsenic in drinking water, however this levels are exceeded in several cities even up to 262.9 µg/L, as it has been previously shown [6]. Chronic arsenic exposure is associated with skin diseases and diabetes mellitus in humans [7, 8]. The potential to develop cancer is also increased in liver, lung, bladder and kidney due to ingested inorganic arsenic in drinking water [9]. It has been previously shown that selenium can antagonize arsenic toxicity in several organs in humans and animals in previous studies from other lab groups and ours [10-12]. We did not observe signs of toxicity with any of the treatments, and therefore we did not find evidence about a reduction of arsenic toxicity when animals were treated with antioxidants.
References
[1] L. J et al., “Transplacental arsenic plus postnatal 12-O-teradecanoyl phorbol-13-acetate exposures associated with hepatocarcinogenesis induce similar aberrant gene expression patterns in male and female mouse liver,” Toxicol. Appl. Pharmacol., vol. 213, no. 3, pp. 216–223, Jun. 2006.
[2] W. MP, W. JM, and D. BA, “Induction of tumors of the liver, lung, ovary and adrenal in adult mice after brief maternal gestational exposure to inorganic arsenic: promotional effects of postnatal phorbol ester exposure on hepatic and pulmonary, but not dermal cancers,” Carcinogenesis, vol. 25, no. 1, pp. 133–141, Jan. 2004.
[3] L. AL et al., “Gender and age differences in the metabolism of inorganic arsenic in a highly exposed population in Bangladesh,” Environ. Res., vol. 106, no. 1, pp. 110–120, Jan. 2008.
[4] L. Chi, X. Bian, B. Gao, H. Ru, P. Tu, and K. Lu, “Sex-Specific Effects of Arsenic Exposure on the Trajectory and Function of the Gut Microbiome,” Chem. Res. Toxicol., vol. 29, no. 6, p. 949, Jun. 2016.
[5] T.-S. L et al., “Sex differences in the reduction of arsenic methylation capacity as a function of urinary total and inorganic arsenic in Mexican children,” Environ. Res., vol. 151, pp. 38–43, Nov. 2016.
[6] R. Hurtado-Jiménez and J. L. Gardea-Torresdey, “Arsenic in drinking water in the Los Altos de Jalisco region of Mexico,” Rev Panam Salud Publica;20(4),oct. 2006, vol. 20, no. 4, 2006.
[7] C. ME, A. A, A. M, and B. E, “Chronic arsenic poisoning in the north of Mexico,” Hum. Toxicol., vol. 2, no. 1, pp. 121–133, 1983.
[8] L. M. Del Razo et al., “Exposure to arsenic in drinking water is associated with increased prevalence of diabetes: a cross-sectional study in the Zimapán and Lagunera regions in Mexico,” Environ. Heal. 2011 101, vol. 10, no. 1, pp. 1–11, Aug. 2011.
[9] C. J. Chen, C. W. Chen, M. M. Wu, and T. L. Kuo, “Cancer potential in liver, lung, bladder and kidney due to ingested inorganic arsenic in drinking water.,” Br. J. Cancer, vol. 66, no. 5, p. 888, 1992.
[10] I. Zwolak, “The Role of Selenium in Arsenic and Cadmium Toxicity: an Updated Review of Scientific Literature,” Biol. Trace Elem. Res., vol. 193, no. 1, p. 44, Jan. 2020.
[11] A. Sampayo-Reyes et al., “Tocopherol and selenite modulate the transplacental effects induced by sodium arsenite in hamsters,” Reprod. Toxicol., vol. 74, pp. 204–211, 2017.
[12] A. Aguirre-Vázquez et al., “Selenite restores Pax6 expression in neuronal cells of chronically arsenic-exposed Golden Syrian hamsters.,” Acta Biochim. Pol., vol. 64, no. 4, pp. 635–639, Dec. 2017.
Reviewer 3 Report
In the manuscript titled “The downregulation of the Stat3 gene induced by selenium, in the liver of chronically exposed Syrian golden hamsters, is not modified by arsenic co-exposure” the authors evaluate the effect of arsenic exposure on the expression of the Stat3 and Psmd10 genes, in the livers of Syrian golden hamsters.
There is still a lot to be determined in this work. I believe that a lot of things need to be clarified and existing things need to be argued much better. Possibly add some more data to support the results obtained. Therefore I think that I can reconsider the possibility of publication after a deep major revision.
- The title is dispersive and too long
- The abstract is too long. The abstract must be 200 words. In rewriting abstract make this more clear. The abstract needs of English improvement.
- When authors mention STAT3 for the first time they have to write it in full, as well as for many other things for which there is only an acronym.
- genes must be written in italics
- Explain more precisely why just these two genes were used
- Why a trend towards a decreased weight in selenite-exposed animals was observed, coinciding with the treatments associated with hepatic lymphocytosis?
- After RNA extraction, do the authors not remove genomic DNA prior to retrotranscription?
- Lines 100-102 “The assessment of arsenic species in 100 urine samples, performed as previously reported [25]” ….the method must be declared in materials and method
- Based on what were the doses of selenite and antioxidants used chosen?
- metal accumulation was done by ICP-MS?
- the correlation between arsenic and expression of tested genes needs to be better argued
- explain the reasons for the levels of the different arsenic species found in urine
- The discussion is chaotic. You can't follow a logical thread. You have to rewrite it with a logical consecuzio
- There is still a lot to be determined in this work. For example, the connection between the alterations of the genes considered and the alterations observed with immunohistochemistry should be better argued
- The conclusion of the work is: selenite down-regulates Stat3 expression and provokes lymphocytosis in the liver. The authors should better explain why this happens. Authors should describe the mechanisms underlying this results or at least make hypotheses. To this end, I suggest that the authors read and cite the following work:10.3390/antiox8070220.
- liver injury induced by different causes including viruses, chemicals and drugs can be protected by different medicinal plants. To this end, I suggest that the authors read and cite the following work: 1080/14786419.2018.1543686.
Author Response
Comment: In the manuscript titled “The downregulation of the Stat3 gene induced by selenium, in the liver of chronically exposed Syrian golden hamsters, is not modified by arsenic co-exposure” the authors evaluate the effect of arsenic exposure on the expression of the Stat3 and Psmd10 genes, in the livers of Syrian golden hamsters.
There is still a lot to be determined in this work. I believe that a lot of things need to be clarified and existing things need to be argued much better. Possibly add some more data to support the results obtained. Therefore I think that I can reconsider the possibility of publication after a deep major revision.
The title is dispersive and too long
Response: The title has been rewritten and shortened.
Comment: The abstract is too long. The abstract must be 200 words. In rewriting abstract make this more clear. The abstract needs of English improvement.
Response: We have rewritten the abstract section and doublecheck the English language.
Comment: When authors mention STAT3 for the first time they have to write it in full, as well as for many other things for which there is only an acronym.
Response: The acronyms have been defined when they are mentioned for first time.
Comment: genes must be written in italics
Response: Gene names have now been written in italics.
Comment: Explain more precisely why just these two genes were used
Response: Thank you for your comment, further information on these genes has been added in the Introduction section to understand the importance in cancer development.
Comment: Why a trend towards a decreased weight in selenite-exposed animals was observed, coinciding with the treatments associated with hepatic lymphocytosis?
Response: Although lymphocytosis findings have been previously observed by selenite treatment (Jacevic´s work), we have no evidence that this finding is related to weight loss, since the steatosis observed in this group was also observed in the control groups; so, steatosis may be due to the diet used. This is now discussed in the Discussion section
Comment: After RNA extraction, do the authors not remove genomic DNA prior to retrotranscription?
Response: DNA was not removed, because primers have been designed on different exons for each gene, where introns are in the middle; this is a current practice in molecular biology labs to avoid the amplification from genomic DNA.
Comment: Lines 100-102 “The assessment of arsenic species in 100 urine samples, performed as previously reported [25]” the method must be declared in materials and method
Response: Thank you for your comment. Information about the method for arsenic species evaluation has been added in the Materials and Methods section.
Comment: Based on what were the doses of selenite and antioxidants used chosen?
Response: As we have previously commented, the doses were selected based on the literature, where it was shown that the doses do not cause toxicity in hamsters. We have added this explanation in Results and Discussion sections.
Comment: metal accumulation was done by ICP-MS?
Response: arsenic detection was performed by the hydride-generation atomic absorption spectrometry (HG-AAS) method, now explained in more detail in Materials and Methods section.
Comment: the correlation between arsenic and expression of tested genes needs to be better argued
Response: We have rewritten relationship between arsenic and the evaluated genes.
Comment: explain the reasons for the levels of the different arsenic species found in urine
Response: We have added more information about arsenic metabolism in the Discussion section to understand why the different arsenic species were evaluated.
Comment: The discussion is chaotic. You can’t follow a logical thread. You have to rewrite it with a logical consecuzio
Response: Thank you for your comment. Discussion has been reorganized to a better comprehension.
Comment: There is still a lot to be determined in this work. For example, the connection between the alterations of the genes considered and the alterations observed with immunohistochemistry should be better argued
Response: We agree reviewer’s comment. It is difficult to establish a relationship between histological changes with gene expression at this point. More studies are needed to know if there is an association between both events.
Comment: The conclusion of the work is: selenite down-regulates Stat3 expression and provokes lymphocytosis in the liver. The authors should better explain why this happens. Authors should describe the mechanisms underlying these results or at least make hypotheses. To this end, I suggest that the authors read and cite the following work:10.3390/antiox8070220.
Response: We have added a hypothesis in the conclusion section.
Comment: liver injury induced by different causes including viruses, chemicals and drugs can be protected by different medicinal plants. To this end, I suggest that the authors read and cite the following work: 1080/14786419.2018.1543686.
Response: Done.
Round 2
Reviewer 1 Report
The corrections and changes made to improve the manuscript were appropriate.
Reviewer 3 Report
Accept in present form